# Quality of Life and Well-Being of Older Adults in Nursing Homes: Systematic Review

Antonia Rodríguez-Martínez [1], Yolanda María De-la-Fuente-Robles [1], María del Carmen Martín-Cano [1] and Juan José Jiménez-Delgado [2,*]

1   Department of Psychology, University of Jaén, 23071 Jaén, Spain; armartin@ujaen.es (A.R.-M.);
    ymfuente@ujaen.es (Y.M.D.-l.-F.-R.); mmcano@ujaen.es (M.d.C.M.-C.)
2   Department of Computer Science, University of Jaén, 23071 Jaén, Spain
*   Correspondence: juanjo@ujaen.es

**Abstract:** The Quality of Life (QoL) of older adults in nursing homes depends on multiple factors. It is necessary to discover the dimensions of QoL, and to obtain an integrating model, analyzing their relationships. With this aim, an exhaustive systematic literature review has been conducted in this area over the last decade. The research question has been to obtain the advances on the key factors influencing the QoL and well-being of older adults living in nursing homes in the last decade. Multiple databases such as Scopus, Web of Science, Wiley Online Library, PubMed, ProQuest, EBSCOhost, and Emerald were used. This review was reported according to the Preferred Reporting Items for Systematic Reviews and Meta-Analysis (PRISMA) guidelines. "Thematic Synthesis Analysis" was used to analyze the studies. The CADIMA web tool was used to conduct the systematic review. The quality of the studies was assessed. The findings were summarized, obtaining a classification of the relevant studies: models or scales for QoL; vision and perception of the QoL of the stakeholders; and determination of QoL through factors (relationship between factors and predictive factors). The results not only evidence the need for further research into this topic, but also the need for an integrative model of QoL, personalized and adapted both to the residents and the nursing home.

**Keywords:** nursing home; older adult; quality of life; relationship; well-being




## 1. Introduction

The present study focuses on the need to conduct research to establish scales and measurements of a concept that is multifaceted and subjective, and should be adapted to the unique conditions of vulnerability of older adults who, when facing this new stage, experience impairment in their daily functioning, independence and Quality of Life (QoL). These circumstances are what lead the older adults to reside in nursing homes (NH) and differentiate their lives from those living in other environments.

QoL refers to the satisfaction and well-being (WB) that individuals experience in their daily lives. In the context of NH, QoL is a key factor in the care and support of residents, and is considered an important measure of the effectiveness of residential care.

Residential care facilities include NH or long-term care residences. NHs are public or private residential facilities that provide a high level of long-term personal or nursing care for persons (such as the aged or the chronically ill) who are unable to care for themselves properly. Typically, older adults require assistance with their daily activities and receive care and support in NHs. In these facilities, QoL is a significant consideration to ensure residents live safely, comfortably, and in an enriching environment.

However, using a generic measure of QoL or one adapted for older adults is not suitable for older adults residing in NHs, given their unique situation of dependency or specific needs that must be measured appropriately. This is the reason why this study is proposed, with the aim of finding evidence from studies that address the QoL of older adults residing in NH, in order to help identify an appropriate QoL model for this population.

The World Health Organization (WHO; WHOQOL Group (1998)) defines QoL as a multidimensional and subjective concept. It is described as "the individual's perception of his or her position in life in the context of the culture and value system in which he or she lives and in relation to his or her goals, expectations, standards, and concerns". We can include WB as a dimension related to QoL in the research question of this study. WB exists in two dimensions, subjective and objective, as defined by the World Health Organization. Regional Office for Europe (2013): "It comprises an individual's experience of their life as well as a comparison of life circumstances with social norms and values".

It is necessary to develop measurement instruments that reflect the multidimensionality of QoL. Towards this aim, several scales have been developed over the years. In its effort to adopt a reliable measure, the WHOQOL Group designed the WHOQOL-BREF as a generic measure of QoL, which also presents an excellent conceptual and operational structure, psychometric performance and reliability, as well as cultural and language adaptation. Various studies have been conducted to establish reliable measurement instruments for investigating and evaluating QoL in the population. However, studies reflecting valid and reliable QoL indicators in NH are still scarce.

Some questionaries or scales for QoL have been adapted and used for NH residents, specific diseases, or defined geographical environments and locations. An example of the latter is the OPQoL-brief (Bowling et al. (2013)) adapted by Haugan et al. (2020). Other scales, such as the QUALIDEM for measuring QoL for residents with dementia and the Psychosocial QoL Domains questionnaire (Kane et al. (2003)) for residents without dementia, have also been used. However, there is a clear need for higher-quality studies that assess a wider range of measurement properties (Aspden et al. (2014); Li et al. (2021)).

The authors and research analyzed in this regard agree on the necessity for greater methodological quality and rigor, as well as the development of specific research that considers a broader range of indicators to contribute to the development of strategies aimed at improving the QoL of older adults residing in NH, rather than adaptations focused on specific diseases, impairments, timeframes, or geographic locations (Aspden et al. (2014); Haugan et al. (2020); Laybourne et al. (2021)). The transformation of NHs into a person-centered approach, known as NH culture change, is a complex and multifaceted task that should be achieved in the development of QoL models for NHs in the near future (Duan et al. (2021)).

Therefore, it is crucial to obtain useful and reliable measurement instruments for assessing QoL and WB in NHs. With this aim, we pose the following research question: What is the state of research in the last decade regarding key factors influencing the QoL and WB of older adults living in NHs? From now on, we understand that QoL includes WB, differentiating them as necessary. This research question will help identify the relevant aspects, measures, and elements that affect or determine the QoL of older adults in NHs.

The approach taken in this study is similar to previous studies (Lee et al. (2009)) that analyzed and compared different studies on QoL in previous years (from 1994 to 2008). The answer to the research question may help modify the methodology of the interventions with older adults residing in NHs.

## 2. Materials and Methods

### 2.1. Protocol

A qualitative systematic literature review (Grant and Booth (2009)) was conducted to obtain key factors for defining QoL and WB in older adults living in NH. This review was reported according to the Preferred Reporting Items for Systematic Reviews and Meta-Analysis (PRISMA) guidelines (Moher et al. (2009); Page et al. (2021)).

The CADIMA web tool (Kohl et al. (2018)) was used to conduct the systematic review. This tool ensures an automated allocation of records during the process and assists the authors in the question formulation, protocol development, duplicate checking, automated allocation of records during the screening process, study selection, critical appraisal, and documentation.

*2.2. Search Strategy*

The search strategy was developed in collaboration with the Social Work area of the University, which has expertise in NH management for older adults. The following databases were searched in January 2022: Scopus, Web of Science, Wiley Online Library, PubMed, ProQuest, EBSCOhost, and Emerald. The search included other databases such as Medline, APA PsycInfo, Health & Medical Collection, and Nursing & Allied Health Premium, among others (32 databases accessible through ProQuest), and MedLine, CINAHL Complete, and Global Health, among others (all accessible through EBSCOhost). Table 1 displays the relevant set of databases used and the number of studies found.

**Table 1.** Results of literature search and databases used (accessed on 12 January 2022).

| Search String | Database or Further Sources (Results) |
|---|---|
| TITLE(("nursing home" OR "care home" OR "retirement home" OR "old's people home" OR "home for the elderly" OR "residency for the elderly" OR "residential care") AND ("quality of life" OR "life quality" OR "well-being")) AND (LIMIT-TO (SUBJAREA, "MEDI") OR LIMIT-TO (SUBJAREA, "NURS") OR LIMIT-TO (SUBJAREA, "SOCI") OR LIMIT-TO (SUBJAREA, "PSYC") OR LIMIT-TO (SUBJAREA, "HEAL") OR LIMIT-TO (SUBJAREA, "MULT")) AND (PUBYEAR > 2010 AND (PUBYEAR < 2022)) | Scopus (402) |
| (TI = (((("nursing home" OR "care home" OR "retirement home" OR "old's people home" OR "home for the elderly" OR "residency for the elderly" OR "residential care") AND ("quality of life" OR "life quality" OR "well-being")))) AND PY = (2011–2021) | Web of Science (322) |
| In title: ("nursing home" OR "care home" OR "retirement home" OR "old's people home" OR "home for the elderly" OR "residency for the elderly" OR "residential care") AND ("quality of life" OR "life quality" OR "well-being") Limit: [between 2011 AND 2021] | Wiley online library (48) |
| (("nursing home"[Title] OR "care home"[Title] OR "retirement home"[Title] OR "old's people home"[Title] OR "home for the elderly"[Title] OR "residency for the elderly"[Title] OR "residential care"[Title]) AND ("quality of life"[Title] OR "life quality"[Title] OR "well-being"[Title])) AND ("1 January 2011"[Date-Publication]: "31 December 2021"[Date-Publication]) | PubMed (231) |
| ti("nursing home" OR "care home" OR "retirement home" OR "old's people home" OR "home for the elderly" OR "residency for the elderly" OR "residential care") AND ti("quality of life" OR "life quality" OR "well-being") [exclude "press"] Limit: [between 2011 AND 2021] | ProQuest in all databases (575): Medline (226) APA PsycInfo (132) Health & Medical Collection (122) Nursing & Allied Health Premium (106) Publicly Available Content Database (72) Psychology Database (61) Sociological Abstracts (45) Social Services Abstracts (30) Sociological Abstracts (25) ProQuest Dissertations & Theses Global (19) Others (6) |
| ti("nursing home" OR "care home" OR "retirement home" OR "old's people home" OR "home for the elderly" OR "residency for the elderly" OR "residential care") AND ti("quality of life" OR "life quality" OR "well-being") Limit: [between 2011 AND 2021] | EBSCOhost in all databases (498): Medline (228) CINAHL Complete (207) Global Health (27) CAB Abstract (17) Others (19) |
| ((title:"nursing home") OR (title:"care home") OR (title:"retirement home") OR (title:"old's people home") OR (title:"home for the elderly") OR (title:"residency for the elderly") OR (title:"residential care")) AND ((title:"quality of life") OR (title:"life quality") OR (title:"well-being")) Limit: [between 2011 AND 2021] | Emerald (6) |



Terms related to "nursing home", "older adults", "quality of life", and "well-being", along with their synonym forms, have been used. Initially, the search was conducted on abstracts and titles. Upon analyzing the obtained results, it was determined that studies relevant to the review's topic were contained within the titles. To validate this observation, a 10% sample of the results from the abstract-only searches was screened. After screening these studies, it was determined that all relevant papers according the PICo and exclusion criteria contained the search keywords in their titles, and no relevant study was found without the search keywords in the title. Studies from 2011 to the present were included due to the absence of similar studies during this period.

Table 1 presents the search strategies employed in the databases, with each one tailored to the search engine specifications of the respective database. The search strategy design utilized an adapted version of the PICO tool (Riesenberg and Justice (2014a, 2014b)). The PICo framework (Population, Phenomenon of interest, Context) was employed to formulate the question review (Stern et al. (2014)).

*2.3. Study Selection*

All titles and abstracts obtained in the search were independently screened by two authors, searching for the population, phenomenon of interest, and context proposed. Discrepancies were resolved by a third author. In this step, 89 studies were included for full-text revision based on the criteria established by the authors.

The exclusion criteria comprised the following: (1) studies focusing on interventions, experiences, or cases that lack generalizability and do not encompass dimensions or factors applicable to older adults living in NHs; (2) studies exclusively related to mental or physical illnesses (e.g., QoL studies solely applied to residents with mental disorders or those with limited mobility); (3) studies concerning therapies, treatments, palliative care, or death and mourning; (4) articles that were unavailable in full text or written in a language other than English or the authors' native language.

All non-excluded studies, but with relevant content published in journals or renowned conferences, underwent a thorough full-text review. Such studies included comparative studies (e.g., older adults living in NHs or in their own homes), new or modified scales or questionnaires regarding QoL or WB applied to NH, factors influencing QoL or WB in NH (e.g., dimensions or factors like active aging, healthy living, mental health, social health), and studies examining perceptions or improvements of QoL or WB (e.g., studies on residents' perceptions or feelings, exploring how specific factors can enhance residents' QoL or WB).

*2.4. Data Extraction*

Two authors gathered the study characteristics and main outcome of each study. Prior to the final assessment of all selected studies, relevant information from these studies was extracted. To complete this stage, the authors utilized a standardized data collection form, following the methodological recommendations proposed by Butler et al. (2016). The extracted information from each study included: title, authors, publication data, language, objective, population, sample, methodology, study type, and the main outcomes related to QoL or WB. Any discrepancies were resolved through consensus via a joint review between the authors.

*2.5. Critical Appraisal*

Two authors independently assessed the methodological quality of the included studies by scoring various criteria. Any discrepancies in scoring and rating were resolved by consensus between the authors. Studies meeting the inclusion criteria were assessed for methodological quality using the assessment criteria developed by Kmet et al. (2004). This method is applicable to both quantitative and qualitative studies. For quantitative studies, 14 items were scored depending on the degree to which the specific criteria were met ("yes" = 2, "partial" = 1, "no" = 0). Items not applicable to a particular study design were

marked "n/a" and were excluded from the calculation of the summary score. A summary score was calculated for each paper by summing the total score obtained across the relevant items and dividing by the total possible score (i.e.,: 28 − (number of "n/a" × 2)). Similarly, scores for qualitative studies were calculated based on the scoring of ten items. Assigning "n/a" was not permitted for any of the items, and the summary score for each paper was calculated by summing the total score obtained across the ten items and dividing by 20 (the total possible score). The authors, in mutual agreement, established thresholds to determine the inclusion of studies.

*2.6. Content Synthesis*

A narrative synthesis was used to identify the main findings of the included studies. After the selection of studies, a thematic analysis was conducted using the "Thematic Synthesis" protocol (Thomas and Harden (2008)). This involved organizing the findings in themes and subthemes and abstracting the relevant information regarding the description, dimensions, and factors that contribute to the QoL of older adults living in NH. The data were synthesized describing the aim, methodology, sample, and the main outcomes of each study.

Thematic Synthesis follows a three-stage process: line by line coding of the findings based on their meaning and content; structuring the codes into descriptive themes and hierarchically grouping themes; and developing new analytical themes to gain a deeper understanding. Throughout these stages, a consensus was reached regarding the generation and subdivision of thematic lines. This process was conducted simultaneously and independently by two authors, with any discrepancies resolved through consensus.

## 3. Results

### 3.1. Selection Process

The literature search and study selection are depicted in the PRISMA diagram shown in Figure 1. In the initial search, 2082 documents were obtained from the databases. After removing duplicates, 623 documents remained for the screening phase.

To address the adequacy of the developed criteria, a consistency check was conducted. For this aim, two authors checked the criteria for 10% of the articles (62 studies). The agreement strength was considered "excellent" with a kappa value of 0.90812.

During the screening phase, 534 papers were randomly reviewed by two authors based on the title and abstract to determine their compliance with the PICo criteria. Following this review, 534 papers were excluded. Discrepancies were resolved by a third author.

A total of 89 studies were selected for full-text review. Among them, 68 studies were excluded for various reasons: 8 studies due to the unavailability of full texts, 10 studies published in non-relevant journals or conferences, 38 studies based on the exclusion criteria, and 10 studies in a language other than English or the authors' native language. Among the remaining 23 studies that initially met the defined criteria, 2 studies were systematic reviews on related topics. Eventually, 21 studies were included for content analysis.

### 3.2. Characteristics of the Studies Included in the Content Synthesis

The 21 studies finally included cover the period from 2012 to 2021, and span across 14 different countries on 5 continents. A total of 7 were qualitative research studies, while 14 were quantitative in 16 different journals. A total of 5 of these studies were published in the journal *Quality of Life Research* and the remaining in other relevant publications.

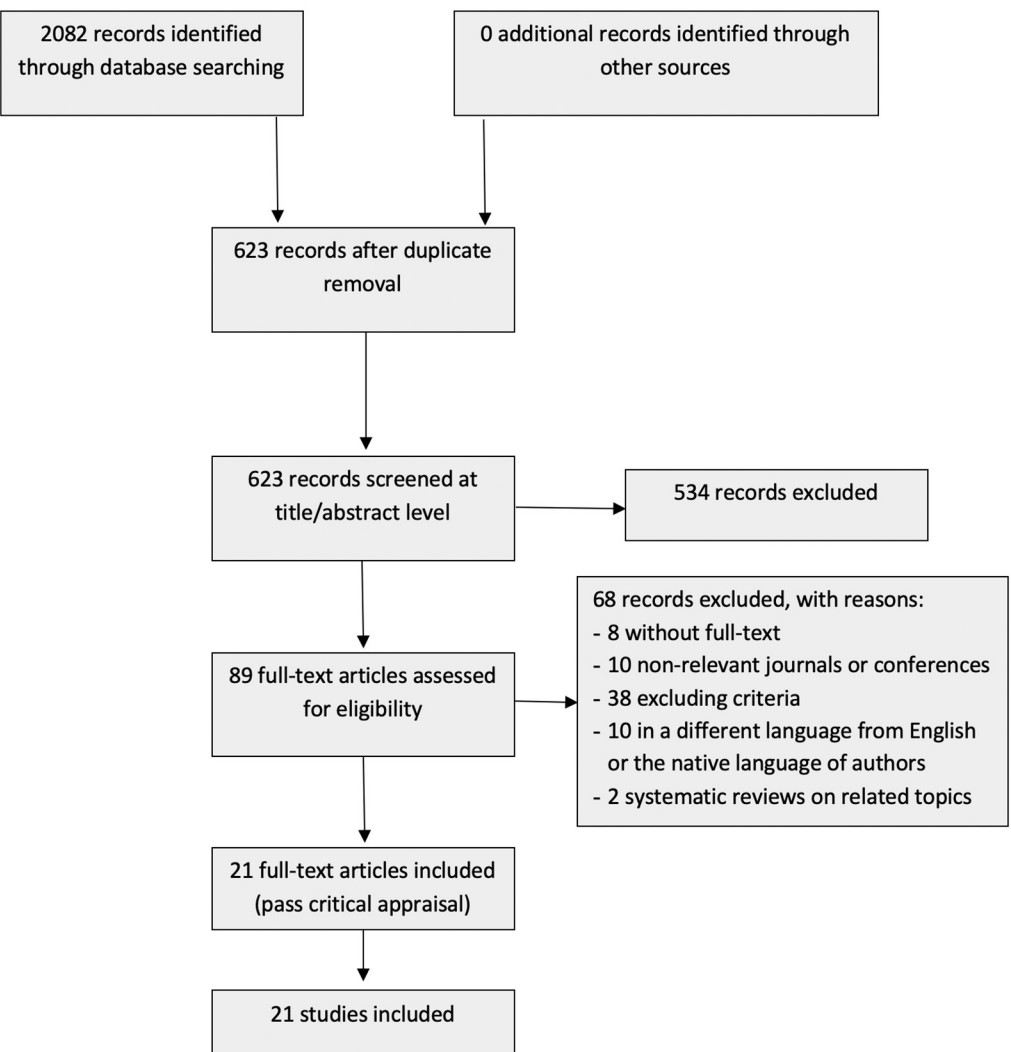

**Figure 1.** Reporting items for Systematic Reviews and Meta-Analysis (PRISMA) flow chart.

### 3.3. Methodological Quality Assessment

A threshold of 75% has been established for the quality of the studies included in the review. All of the studies included in the review surpass this threshold, both quantitative and qualitative studies. The quality percentage for qualitative studies is generally lower, averaging around 78% compared to an average of 90% for quantitative studies.

Item 12 "controlled for confounding" received a lower rating for two quantitative studies (Carcavilla González et al. (2021); Pramesona and Taneepanichskul (2018)), as they did not account for confounding or the dependency between variables. The lowest quality percentage obtained for quantitative studies was 77% (Kloos et al. (2019)). The qualitative studies range from 75% to 80% in terms of quality percentage, with this low-quality percentage being attributed to the absence of a verification procedure to establish credibility (item 8). Additionally, the conclusions are partially supported by the data, and the results are not generalizable (item 9). The quality assessment of the included studies is presented in Table 2 (quantitative) and Table 3 (qualitative).

**Table 2.** Quality assessment criteria scores for quantitative methodologies (Kmet et al. (2004)).

| Question | Carcavilla González et al. (2021) | Godin et al. (2015) | Xu et al. (2019) | Haugan et al. (2020) | Burack et al. (2012) | Duan et al. (2021) | Kloos et al. (2019) | McCabe et al. (2021) | Maenhout et al. (2020) | Nordin et al. (2017) | Pramesona and Taneepanichskul (2018) | Roberts and Ishler (2018) | Scocco and Nassuato (2017) | Wu et al. (2018) |
|---|---|---|---|---|---|---|---|---|---|---|---|---|---|---|
| 1 | 2 | 2 | 2 | 2 | 2 | 2 | 2 | 2 | 2 | 2 | 2 | 2 | 2 | 2 |
| 2 | 1 | 2 | 2 | 2 | 2 | 2 | 1 | 2 | 2 | 1 | 2 | 2 | 1 | 2 |
| 3 | 2 | 2 | 2 | 2 | 2 | 2 | 2 | 2 | 2 | 1 | 1 | 2 | 1 | 1 |
| 4 | 1 | 1 | 2 | 1 | 1 | 2 | 1 | 1 | 2 | 1 | 2 | 2 | 2 | 2 |
| 8 | 2 | 2 | 2 | 2 | 2 | 2 | 1 | 2 | 2 | 2 | 2 | 2 | 2 | 2 |
| 9 | 2 | 2 | 2 | 2 | 1 | 2 | 2 | 2 | 2 | 2 | 2 | 2 | 2 | 2 |
| 10 | 2 | 2 | 2 | 2 | 1 | 2 | 1 | 2 | 2 | 2 | 1 | 2 | 2 | 1 |
| 11 | 2 | 2 | 2 | 2 | 2 | 2 | 2 | 2 | 2 | 2 | 2 | 2 | 2 | 2 |
| 12 | 0 | 2 | 2 | 2 | 1 | 2 | 1 | 2 | 2 | 2 | 0 | 2 | 1 | 2 |
| 13 | 2 | 1 | 2 | 2 | 2 | 2 | 2 | 1 | 2 | 2 | 2 | 2 | 2 | 2 |
| 14 | 2 | 2 | 2 | 2 | 2 | 2 | 2 | 1 | 2 | 2 | 2 | 2 | 2 | 2 |
| Score | 18 | 20 | 22 | 21 | 18 | 22 | 17 | 19 | 22 | 19 | 18 | 22 | 19 | 20 |
| Max. | 22 | 22 | 22 | 22 | 22 | 22 | 22 | 22 | 22 | 22 | 22 | 22 | 22 | 22 |
| % | 82% | 91% | 100% | 95% | 82% | 100% | 77% | 86% | 100% | 86% | 82% | 100% | 86% | 91% |

1 Question/objective sufficiently described? 2 Study design evident and appropriate? 3 Method of subject/comparison group selection or source of information/input variables described and appropriate? 4 Subject (and comparison group, if applicable) characteristics sufficiently described? 8 Outcome and (if applicable) exposure measure(s) well defined and robust to measurement/misclassification bias? Means of assessment reported? 9 Sample size appropriate? 10 Analytic methods described/justified and appropriate? 11 Some estimate of variance is reported for the main results? 12 Controlled for confounding? 13 Results reported in sufficient detail? 14 Conclusions supported by the results? Questions 5, 6 and 7 have been excluded as interventional studies are not included in this systematic review.

**Table 3.** Quality assessment criteria scores for qualitative methodologies (Kmet et al. (2004)).

| Question | Adra et al. (2015) | Adra et al. (2017) | Johs-Artisensi et al. (2020) | Meyer et al. (2019) | Schenk et al. (2013) | van Biljon et al. (2015) | van Biljon and Roos (2015) |
|---|---|---|---|---|---|---|---|
| 1 | 2 | 2 | 2 | 2 | 2 | 2 | 2 |
| 2 | 1 | 2 | 2 | 2 | 1 | 2 | 2 |
| 3 | 2 | 2 | 2 | 2 | 2 | 2 | 2 |
| 4 | 2 | 2 | 1 | 2 | 1 | 1 | 2 |
| 5 | 1 | 1 | 2 | 1 | 2 | 2 | 1 |
| 6 | 2 | 2 | 2 | 2 | 2 | 2 | 2 |
| 7 | 2 | 1 | 2 | 2 | 2 | 2 | 2 |
| 8 | 0 | 0 | 0 | 0 | 0 | 0 | 0 |
| 9 | 1 | 1 | 1 | 1 | 1 | 1 | 1 |
| 10 | 2 | 2 | 2 | 2 | 2 | 2 | 2 |
| Score | 15 | 15 | 16 | 16 | 15 | 16 | 16 |
| Max. | 20 | 20 | 20 | 20 | 20 | 20 | 20 |
| % | 75% | 75% | 80% | 80% | 75% | 80% | 80% |

1 Question/objective sufficiently described? 2 Study design evident and appropriate? 3 Context for the study clear? 4 Connection to a theoretical framework/wider body of knowledge? 5 Sampling strategy described, relevant, and justified? 6 Data collection methods clearly described and systematic? 7 Data analysis clearly described and systematic? 8 Use of verification procedure(s) to establish credibility? 9 Conclusions supported by the results? 10 Reflexivity of the account?

### 3.4. Findings of the Studies

The findings of this review can be classified into three main thematic groups, following a similar approach to Lee et al. (2009) as mentioned in the introduction. Firstly, there are studies that describe or develop specific QoL models or scales for older adults living in NHs. Secondly, there are studies that qualitatively explore stakeholders' perceptions of QoL. Lastly, there are studies that examine the relationships between QoL factors, some of which may serve as predictors of QoL. The aforementioned studies are summarized in Table 4.

**Table 4.** Summary of studies included in the review.

| Authors/Year and Title | Country/Sample | Aim | Classification | Methodology/Design | Main Findings |
|---|---|---|---|---|---|
| Carcavilla González et al. (2021) Development of a subjective quality of life scale in nursing homes for older people (CVS-R) | (Spain) n = 99 (development, 36.4% professionals, 30.3% residents, 33.33% family members) n = 225 (validity, 62% professionals, 23% residents, 14% family members) | To develop and validate a QoL questionnaire for nursing home residents in Spain | QoL and WB models or scales | Literature review. Pilot study. Factorial analysis of principal components, evaluation of the internal consistency for its reliability (Cronbach's alpha coefficient) | A validated and reliable questionnaire to measure QoL in NHs, contemplating the perception of residents, family members, and professionals (with 27 questions and 9 dimensions) |
| Godin et al. (2015) Nursing home resident quality of life: testing for measurement equivalence across resident, family, and staff perspectives | (Canada) n = 319 (residents), n = 397 (family members), n = 862 (staff), n = 23 (nursing homes) | To explore the factor structure of the interRAI self-report nursing home QoL survey and to develop a measure that will allow researchers to compare predictors of QoL across resident, family, and staff perspectives | QoL and WB models or scales | Exploratory and confirmatory factor analysis | A model with a four-factor structure (i.e., care and support, food, autonomy, and activities) across resident, family, and staff perspectives. A tool that researchers can use to compare predictors of QoL |

**Table 4.** *Cont.*

| Authors/Year and Title | Country/Sample | Aim | Classification | Methodology/Design | Main Findings |
|---|---|---|---|---|---|
| Xu et al. (2019) Development of a quality of life questionnaire for nursing home residents in mainland China | (mainland China) n = 176 residents (development) n = 371 residents (validation) | To develop and validate a QoL questionnaire for nursing home residents in mainland China | QoL and WB models or scales | Exploratory and confirmatory factor analysis. Descriptive statistics. Resident interviews, literature reviews, expert panels, and pilot studies. | A nursing home QoL questionnaire with satisfactory reliability and validity, it has 9 domains and 38 items including physical health, food enjoyment, security, environmental comfort, autonomy, meaningful activity, interrelationships, family relationships, and mood |
| Adra et al. (2015) Constructing the meaning of quality of life for residents in care homes in the Lebanon: Perspectives of residents, staff and family | (Lebanon) n = 20 (residents), n = 8 (family caregivers), n = 11 (care staff), across 2 nursing homes | To explore the perspectives of QoL for a sample of older residents, care staff and family caregivers, so far little is known about its meanings from an Arabic cultural perspective and context | Stakeholders' views and perceptions of QoL and WB | Category analysis. Analytical interpretation | Four categories emerged: maintaining family connectedness; engaging in worthwhile activities; maintaining and developing significant relationships; and maintaining and practicing spiritual beliefs |
| Adra et al. (2017) Nursing home quality of life in the Lebanon | (Lebanon) n = 20 (residents), n = 8 (family caregivers), n = 11 (staff), across 2 nursing homes | To explore perceptions, perspectives, and meaning of QoL for a sample of older residents, care staff, and family caregivers in two nursing homes in Lebanon | Stakeholders' views and perceptions of QoL and WB | Category analysis | Three distinct but interrelated properties of QoL emerged from this process: "maintaining self", "maintaining identity", and "maintaining continuity". The dynamics that exist within and between each of these properties provide an indicator about shared and distinct meanings and the implications for care practice |
| Johs-Artisensi et al. (2020) Qualitative analyzes of nursing home residents' quality of life from multiple stakeholders' perspectives | (US) n = 138 (residents), n = 138 (nursing assistants), n = 46 (social workers), n = 46 (activities directors), n = 46 (administrators) | To identify contributory factors to resident QoL, as well as analyze areas of commonality in qualitative responses | Stakeholders' views and perceptions of QoL and WB | Multi-step, inductive approach in order to conduct a thematic analysis to assess patterns of meaning within the datasets from the interview questions | Confirm previous research findings of resident-centered care contributing to residents QoL and distinguish between various stakeholders' perspectives within the nursing home settings. Contributory factors: Activities, Autonomy/Respect, Comfort-ability/Environment, Contributory Service, Emotional Well-Being, Familial Communication, Food/Drink, Quality of Care, Sense of Community, Spirituality/Religion, Staff and Resident Relationships. Staff and Resident Relationships were important to all stakeholders. Greater alignment between nursing assistants and residents. Residents did not rank Quality of Care as one of their top contributory factors, but staff and management all included the |

**Table 4.** *Cont.*

| Authors/Year and Title | Country/Sample | Aim | Classification | Methodology/Design | Main Findings |
|---|---|---|---|---|---|
| Meyer et al. (2019) Questioning the Questionnaire: Methodological Challenges in Measuring Subjective Quality of Life in Nursing Homes Using Cognitive Interviewing Techniques | (Germany) n = 16 residents, across 4 care homes | To analyze how older adults in residential care facilities interpret and process response stimuli received from a questionnaire on subjective QoL. To gain methodological insights into the way a survey instrument on subjective QoL can adequately represent individual ratings, as well as expectations regarding different aspects of QoL | Stakeholders' views and perceptions of QoL and WB | Analysis conducted by consensus using cognitive interviewing techniques within a qualitative validation study | Development of QUISTA assessment tool (QoL in Residential Care). Subjective QoL as a multidimensional construct that includes aspects considered important to nursing home residents. The operationalization of these dimensions as questions based on QoL dimensions reconstructed from their original form (the residents themselves described and reflected on what constitutes dimensions of subjective QoL). The comparison of subjective assessments of QoL aspects (the "actual" state) with personal preferences in relation to the same aspects (the "desired" state) |
| Schenk et al. (2013) Quality of life in nursing homes: Results of a qualitative resident survey | (Germany) n = 42 residents across 8 nursing homes | To identify dimensions of life that nursing home residents perceive as having a particular impact on their overall QoL | Stakeholders' views and perceptions of QoL and WB | The interviews analyzed using the documentary method | Ten central dimensions of subjective QoL were derived from the interview data: social contacts, self-determination and autonomy, privacy, peace and quiet, variety of stimuli and activities, feeling at home, security, health, being kept informed, and meaningful/enjoyable activity. Some of these dimensions are multifaceted and have further subdimensions |
| van Biljon et al. (2015) A Conceptual Model of Quality of Life for Older People in Residential Care Facilities in South Africa | (South Africa) n = 19 residents across 3 nursing homes | How do older adults conceptualize their QoL in residential care facilities in terms of cause-and-effect relations between domains of QoL, based on the six domains deriving from the work of van Biljon and Roos. To obtain a conceptual model of QoL that is reflective of the system dynamics of how older adults construct their QoL in residential care facilities | Stakeholders' views and perceptions of QoL and WB | Interactive Qualitative Analysis | A conceptual model of QoL for older adults in residential care facilities, with 6 domains (spirituality, health, meaningfulness, sense of place, autonomy, and relationships). The domains with the power to reinforce this system were spirituality and autonomy. The domain of spiritually has a cognitive and behavioral transformational ability and has the potential to constructively assist older adults with emotional regulation, health aspects, and also to deal with adversities. Furthermore, autonomy has the potential to give older adults a sense of self-esteem and purpose which will reinforce their ability to live meaningful lives |

**Table 4.** *Cont.*

| Authors/Year and Title | Country/Sample | Aim | Classification | Methodology/Design | Main Findings |
|---|---|---|---|---|---|
| van Biljon and Roos (2015) The nature of quality of life in residential care facilities: The case of White older South Africans | (South Africa) n = 41 residents across 4 nursing homes | To explore QoL as perceived by older adults residing in residential care facilities in South Africa | Stakeholders' views and perceptions of QoL and WB | Narrative reflections on QoL in journals. Qualitative research to explore and describe participants' understanding and interpretation of QoL. Interpretative Phenomenological Analysis to analyze the data | The resident older adult South Africans regard QoL as a spiritually informed worldview of life events, coping with challenges and being mindful of others. The residents perceived QoL to include proximity and quality and reciprocity with others |
| Haugan et al. (2020) Assessing quality of life in older adults: Psychometric properties of the OPQoL-brief questionnaire in a nursing home population | (Norway) n = 188 residents across 27 nursing homes | To test the psychometrical properties of the OPQoL-brief questionnaire among cognitively intact nursing home residents | Relationships between QoL factors | Principal component analysis and confirmative factor analysis | Evidence related to the dimensionality, reliability, and construct validity; all of the items considered interrelated measurement properties. Of the original 13 items, 5 showed low reliability and validity; excluding these items revealed a good model fit for the one-dimensional 8-item measurement model, showing good internal consistency, and validity for these 8 items (anxiety, depression, self-transcendence, meaning-in-life, nurse-patient interaction, and joy-of-life) |
| Burack et al. (2012) What matters most to nursing home elders: Quality of life in the nursing home | (US) n = 62 residents across 3 nursing homes | To determine those components of nursing home QoL that are associated with older adults' satisfaction so as to provide direction in the culture change journey | Predictors of QoL | A cross-sectional study using a survey administered face-to-face. Regression analysis | After accounting for cognitive and physical functioning, among the QoL domains, dignity, spiritual well-being, and food enjoyment remained predictors of overall nursing home satisfaction. Additionally, dignity remained a significant predictor of older adults' satisfaction with staff |
| Duan et al. (2021) The Relationships of Nursing Home Culture Change Practices With Resident Quality of Life and Family Satisfaction: Toward a More Nuanced Understanding | (US) n = 102 administrators | To test the domain-specific relationships of culture change practices with resident QoL and family satisfaction, and to examine the moderating effect of small-home or household models on these relationships | Relationships between QoL factors | Descriptive statistics to describe NH characteristics, culture change domain scores, resident QoL scores, and family satisfaction scores. A linear regression model, separately, for the summary scores of resident QoL and family satisfaction, and their domain scores | Culture change operationalized through physical environment transformation, staff empowerment, staff leadership, and end-of-life care was positively associated with at least one domain of resident QoL and family satisfaction, while staff empowerment had the most extensive effects. Implementing small-home and household models had a buffering effect on the positive relationships between staff empowerment and the outcomes |

**Table 4.** *Cont.*

| Authors/Year and Title | Country/Sample | Aim | Classification | Methodology/Design | Main Findings |
|---|---|---|---|---|---|
| Kloos et al. (2019) Longitudinal Associations of Autonomy, Relatedness, and Competence With the Well-being of Nursing Home Residents | (Netherlands) n = 128 physically frail residents in somatic long-term care units at 4 nursing homes | To test the longitudinal relations of the satisfaction of three basic psychological needs (satisfying nursing home residents' needs for autonomy, relatedness, and competence) to the subjective well-being of nursing home residents and to determine whether a balance among the satisfaction of the three needs is important for well-being | Relationships between WB factors | Correlations between subscales. Hierarchical multiple regression analysis | All three needs (autonomy, relatedness, and competence) were related to both well-being measures over time, although autonomy had the strongest relationships. Only autonomy and competence were uniquely associated with depressive feelings, and only autonomy was uniquely associated with life satisfaction. The need satisfaction balance score was related to well-being independent of the autonomy and relatedness scores |
| McCabe et al. (2021) How Important Are Choice, Autonomy, and Relationships in Predicting the Quality of Life of Nursing Home Residents? | (Australia) n = 604 residents across 33 nursing homes | To evaluate the contribution of resident choice, as well as the staff–resident relationship, to promoting resident QoL | Predictors of QoL | Hierarchical regression | Two of the four predictor variables (resident choice over socializing and the staff–resident relationship) significantly contributed to resident QoL |
| Maenhout et al. (2020) The relationship between quality of life in a nursing home and personal, organizational, activity-related factors and social satisfaction: a cross-sectional study with multiple linear regression analyses | (Belgium) n = 171 cognitively healthy residents across 73 nursing homes | To investigate QoL in nursing home residents and the relationship with personal, organizational, activity-related factors, and social satisfaction | Relationships between QoL factors | Cross sectional survey. Multiple linear regression (forward stepwise selection) | Results suggest that a higher QoL in nursing homes can be pursued by strategies to prevent depression and to improve nursing home residents' subjective perception of health (e.g., offering good care) and social networking |
| Nordin et al. (2017) The association between the physical environment and the well-being of older people in residential care facilities: A multilevel analysis | (Sweden) n = 200 residents across 20 nursing homes | To investigate the associations between the quality of the physical environment and the psychological and social well-being of older adults living in residential care facilities | Relationships between WB factors | A cross-sectional survey of care facilities. Multilevel analysis | Cognitive support in the physical environment was associated with residents' social well-being, after controlling for independence and perceived care quality. No significant association was found between the physical environment and residents' psychological well-being |

**Table 4.** *Cont.*

| Authors/Year and Title | Country/Sample | Aim | Classification | Methodology/Design | Main Findings |
|---|---|---|---|---|---|
| Pramesona and Taneepanichskul (2018) Factors influencing the quality of life among Indonesian elderly: A nursing home-based cross-sectional survey | (Indonesia) n = 181 residents across 3 nursing homes | To examine the level of QoL and factors influencing QoL amongst older adult NH residents in Indonesia | Predictors of QoL | Descriptive statistics. Multivariate linear regression | Perceived adequacy of care and reason for living in an NH were highlighted as predictors of QoL amongst older adult NH residents |
| Roberts and Ishler (2018) Family Involvement in the Nursing Home and Perceived Resident Quality of Life | (US) n = 14,979 family members across 839 nursing homes | To study the relationship between family involvement and family perceptions of nursing home residents' QoL | Predictors of QoL | Hierarchical linear modelling was used to examine the association between family involvement and other predictors with perceived resident QoL | Although most of the variability in family member perceptions of resident QoL was observed at the individual level (residents and families), characteristics of the facilities were also significantly associated with perceived resident QoL. Family involvement was a strong predictor of perceived resident QoL: families who visited frequently and provided more help with personal care perceived lower resident QoL, while those who communicated frequently with facility staff had higher perceptions of resident QoL. The negative association between helping with more personal care and perceiving lower resident QoL was attenuated when family members communicated more regularly with facility staff. However, as family member age increased, the positive association between communication with facility staff and resident QoL diminished. Family members who are spouses, older, non-White, and highly educated perceived resident QoL as lower |

**Table 4.** *Cont.*

| Authors/Year and Title | Country/Sample | Aim | Classification | Methodology/Design | Main Findings |
|---|---|---|---|---|---|
| Scocco and Nassuato (2017) The role of social relationships among elderly community-dwelling and nursing-home residents: findings from a quality of life study | (Italy) n = 207 older adults (n = 135 community-dwelling residents, n = 72 nursing home residents across 2 nursing homes) | To compare World Health Organization QoL brief version (WHOQOL-BREF) scores of the community of older adult dwelling residents and nursing home residents, based on the assumption that QoL, particularly social relationships, may be perceived differently according to residential setting | Relationships between QoL factors | Linear regression model. Logistic regression model | Depressive symptoms correlated with low scores in all WHOQOL-BREF domains. The variables that correlated with living conditions in a nursing home were older age, male gender, lower physical domain scores, and higher social relationship scores |
| Wu et al. (2018) Association between social support and health-related quality of life among Chinese rural elders in nursing homes: the mediating role of resilience | (China) n = 205 residents across 5 nursing homes | To confirm the relationship between social support and health-related QoL (HRQOL) among rural Chinese older adults in nursing homes, and to examine the mediating role of resilience in the impact of social support on HRQOL | Relationships between QoL factors | Cross sectional study. Statistical analysis. Correlation matrix, with Pearson's coefficients for continuous variables or Spearman's coefficients for nominal and ordinal variables. Mediation analysis | Social support was positively related to QoL among older adults. In addition, the mediating role of resilience in the relationship between social support and QoL is confirmed |

Thematic lines emerged from three different themes as follows: (1) QoL and WB models or scales: when a study uses a new questionnaire or integrates measures from different sources; (2) stakeholders' views and perceptions of QoL and WB: when a study seeks to gain a deeper understanding of the concept of QoL in residents regarding other stakeholders; and (3) determination of QoL and WB through factors and their relationships: when the primary focus of the study is to synthesize pertinent information or examine the influence of factors on each other.

### 3.4.1. QoL and WB Models or Scales

The measurement of QoL is not a totally standardized process. The measures developed depend on many factors, including regional or country-specific factors, cultural considerations, societal influences, and diverse understandings of the QoL concept. In the case of QoL of older adults living in NHs, there are additional factors and specific circumstances that require adapting existing scales or developing new measurement methods.

Among the 21 studies, 3 studies designed their own questionnaires to measure QoL in NHs. Details about these proposed models are provided in Table 4.

One study utilized Ryff's model of psychological well-being (Ryff (1991)) along with a Person-Centered Care approach to improve the quality of care for older adults, irrespective of their overall health (Carcavilla González et al. (2021)). A multiple-perspective evaluation of QoL is achievable through the design of a QoL scale that incorporates not only objec-

tive indicators of NHs but also subjective aspects based on the perceptions of the users, their relatives, and professionals in the facility (Carcavilla González et al. (2021); Godin et al. (2015)).

QoL is a subjective concept that encompasses dimensions beyond the health or functional status of residents. This subjectivity allows for the development of QoL models that examine the perspectives of residents, family members, and professionals. The perception of family members can influence the appreciation of residents, as well as the allocation of resources and care provided by professionals. Both family members' and professionals' perceptions play a role in shaping residents' daily care and activities (Godin et al. (2015)).

Furthermore, conducting literature reviews on different QoL dimensions and utilizing qualitative studies can help identify new domains and items related to QoL. Interviews, expert panels, and pilot studies contribute to the development of specific questionnaires, employing exploratory and confirmatory factor analyses to test their validity regarding the residents' QoL. By focusing on a specific locality population, important domains can be identified from the residents' perspective, serving as a foundation for the development of a tailored QoL questionnaire. Combining this information with other questionnaires such as health-related QoL and other variables enables the construction of a robust and specific measure for a specific population in a particular location, such as mainland China (Xu et al. (2019)).

### 3.4.2. Stakeholders' Views and Perceptions of QoL and WB

Residents' perceptions of QoL should be prioritized, although other external factors are also important. The perspectives of various stakeholders were considered in 7 out of the 21 studies. These studies utilized interviews to explore new dimensions or domains of QoL. Some of these studies focused on dimensions such as spirituality, proximity to peers, and reciprocity (van Biljon and Roos (2015)). According to the aforementioned study, QoL in this context encompasses six domains: spirituality, health, meaningfulness, sense of place, autonomy, and relationships. Furthermore, another study by the same authors revealed that autonomy enables older adults to develop a sense of self-esteem and purpose, which enhances their ability to lead meaningful lives (van Biljon et al. (2015)).

If resident interviews are conducted, it is often necessary to customize the questions to suit the residents' specific circumstances. While certain dimensions such as spirituality, proximity to peers, and reciprocity may not be significant in other contexts, these perceptions play a crucial role in shaping the QoL of residents in South Africa (van Biljon and Roos (2015)). According to the aforementioned study, QoL in this context encompasses six domains: spirituality, health, meaningfulness, sense of place, autonomy, and relationships. Through an Interactive Qualitative Analysis (IQA), it was determined that the belief system and spiritual worldview are important components of QoL. The spiritual domain has the potential to positively influence cognitive and behavioral transformation, emotional regulation, aspects of health, and coping with adversity among older adults. Additionally, autonomy provides older adults with a sense of self-esteem and purpose, thereby strengthening their ability to lead meaningful lives (van Biljon and Roos (2015)).

Maintaining family connections, engaging in valued activities, fostering meaningful relationships, and practicing spiritual beliefs are significant aspects within a specific setting. These factors, combined with the perspectives of various stakeholders, contribute to shaping a tailored definition of QoL within a particular territorial context (Adra et al. (2015)). The definition of QoL has evolved through diverse approaches over time, influenced by different stakeholders (Adra et al. (2015, 2017); Johs-Artisensi et al. (2020)). Achieving adequate QoL for residents requires aligning the perspectives of different groups. Johs-Artisensi et al. (2020) propose a better understanding of residents' subjective perception of QoL to identify factors that contribute to a higher QoL and to recognize areas of intergroup agreement or disagreement.

The construct of QoL encompasses multiple dimensions that differentiate between objective and subjective aspects. Objective aspects pertain to the quality of conditions

and standards set by experts, while subjective aspects relate to the quality of personal experiences, such as satisfaction with conditions and WB, which are assessed based on individual standards and measured through self-reports. However, individual needs are often overlooked in this context, highlighting the necessity of developing measures for subjective QoL in NHs. Schenk et al. (2013) identified ten core dimensions of subjective QoL, including social contacts, self-determination and autonomy, privacy, peace and quiet, variety of stimuli and activities, feeling at home, security, health, being kept informed, and meaningful/enjoyable activity. Some of these dimensions are multifaceted and have further subdimensions.

An analysis of data derived from traditional NH satisfaction surveys indicates a potential positive bias in responses. It is crucial to recognize that residents are a vulnerable and sometimes challenging group to survey. The study conducted by Meyer et al. (2019) explores how residents interpret and process response stimuli received from a subjective QoL questionnaire, thereby contributing methodological insights into adequately representing individual scores and expectations regarding different aspects of QoL.

### 3.4.3. Determination of QoL and WB through Factors and Their Relationships

This category includes studies that establish relationships between factors influencing QoL and WB, as well as those that determine predictive values of QoL. These studies aim to investigate whether certain circumstances or the relationships between variables impact the QoL of NH residents, summarizing or predicting their QoL. Out of the 21 studies analyzed, 11 focused on these factors.

By establishing QoL dimensions, it becomes possible to define relationships between these dimensions or variables. Of the 21 studies, 7 pursued this line of research. The specific circumstances of each NH, the residents, the region in which they are located, and other factors contribute to variations in the QoL measure and the factors that influence it. Therefore, different dimensions with varying degrees of importance are required depending on the specific case.

According to Maenhout et al. (2020), dimensions related to personal, organizational, or activity-related factors, as well as social satisfaction, can be used to determine their relationship with residents' QoL. Improving residents' subjective perception of health, preventing depression, and enhancing their social networks can contribute to an overall improvement in their QoL.

The use of non-specific QoL measurement instruments for older adults living in NHs can provide valuable insights into the factors that influence this specific population. A comparison can be made between their QoL and that of older adults living in the community. According to Scocco and Nassuato (2017), socialization opportunities in NHs can improve residents' perception of QoL compared to community-dwelling older adults. This study also found that depressive symptoms affect QoL in both populations.

Additionally, the OPQoL-brief questionnaire (Bowling et al. (2013)) in its Norwegian version has demonstrated good psychometric properties for measuring QoL in residents. However, Haugan et al. (2020) suggest that not all items of this questionnaire are suitable for this population. By excluding certain items, such as anxiety, depression, self-transcendence, meaning in life, nurse–patient interaction, and joy of life, a more accurate model for measuring QoL with good internal consistency and validity can be obtained.

Another specific dimension of QoL is related to health factors, known as Health-Related Quality of Life (HRQOL). The study by Wu et al. (2018) provides evidence that social support is associated with this dimension of QoL, with resilience playing a mediating role in the impact of social support. Resilience partially mediates the relationship between social support and HRQOL.

Determining the relationship between NH culture change and QoL is an emerging area of study (Burack et al. (2012); Duan et al. (2021)). According to Duan et al. (2021), culture change practices in the physical environment, staff empowerment, staff leadership, and end-of-life care are positively associated with a specific domain of resident QoL and

family satisfaction. It should be noted that higher-quality physical environments may support older adults with increased frailty and promote their WB. However, no significant association has been found between the physical environment and residents' psychological WB. On the other hand, cognitive support in the physical environment is associated with residents' social WB, even after accounting for independence and perceived quality of care (Nordin et al. (2017)). Psychological theories of WB support the cultural change movement by identifying resident-level factors that contribute to QoL. Meeting NH residents' needs for autonomy, relatedness, and competence can enhance their WB. These three needs have been found to be related to measures of WB over time, with autonomy showing the strongest relationship (Kloos et al. (2019)).

Certain factors can be predictive of higher or lower QoL. Out of the 21 studies in the literature, 4 took this approach. In the study conducted by McCabe et al. (2021), it was found that residents' choices regarding various aspects of their lives (such as food and leisure choice, socialization, and care), and the relationship between professionals and residents to foster autonomy and promote improvement in residents' QoL, are predictors of residents' QoL. However, it should be noted that only residents' choices regarding socialization and the professional–resident relationship were found to be significant contributors to QoL. Additionally, autonomy and social relationships were found to have a positive impact on residents' QoL.

According to Pramesona and Taneepanichskul (2018), residents' perceptions of the adequacy of care and having a reason for living in NHs are predictors of QoL. It is important to consider the perception of QoL not only among residents but also among other stakeholders, particularly for residents with dementia, low cognition, or severe physical impairments who may have difficulty expressing their own QoL. Additionally, the characteristics of the residential facility are significantly associated with residents' perceived QoL, as noted by Roberts and Ishler (2018). Family involvement is a strong predictor of QoL, with families who visit frequently and assist in resident care perceiving better QoL, and families who have more frequent communication with facility staff perceiving higher resident QoL.

In the context of culture change practices in NH, as studied by Burack et al. (2012), using a QoL scale with various domains such as autonomy, dignity, food enjoyment, functional competence, individuality, meaningful activity, physical comfort, privacy, relationships, safety, and spiritual well-being, it can be suggested that dignity, spiritual well-being, and food enjoyment are predictors of overall NH satisfaction. Furthermore, dignity remains a significant predictor of older adults' satisfaction with staff.

## 4. Discussion

Of the 21 significant papers that were found, 3 studies establish new models or scales for measuring QoL specifically designed for NH settings. Additionally, 7 studies provide a qualitative description of the perception of QoL in NHs, while 11 studies identify factors related to QoL in NHs, examine their relationships, and, in some cases, explore variables that predict QoL in this context.

The perception of QoL is determined by various dimensions that are highly dependent on the specific circumstances of the residence, as well as other economic, cultural, religious, and other factors. It is important to note that the findings from these studies are valid within the geographic areas mentioned and may not be directly applicable in other contexts. Therefore, further research is needed to determine the factors that are relevant to the QoL of NH residents in different countries (McCabe et al. (2021)).

Based on the aforementioned findings, this systematic review of the literature has identified studies that contribute to the development of new QoL scales. However, the number of studies conducted in this area during the past decade is relatively limited, indicating a potential lack of attention and a need for further development in this field of study. These studies emphasize that QoL is a multidimensional construct encompassing both objective and subjective dimensions, observed from various perspectives and disciplines

such as institutions, gerontology, healthcare, and psychology, among others. These studies highlight the significance of assessing different domains of QoL, including physical health (Xu et al. (2019)), psychological WB (Carcavilla González et al. (2021)), care and support (Godin et al. (2015)), and environmental factors (Xu et al. (2019)). Moreover, they provide evidence regarding the validity and reliability of these scales in diverse geographic areas.

Most of the studies focused on understanding QoL in NHs aim to identify the different dimensions that constitute the QoL construct through the perspectives of various stakeholders. Their objective is to conceptualize the QoL of NH residents. This group of studies primarily seeks to explore the subjective experiences and perceptions of QoL among residents, family members, and staff in NHs. Within this group, seven studies provided a qualitative description of QoL perceptions, emphasizing the importance of social relationships (Adra et al. (2015); Johs-Artisensi et al. (2020)), autonomy (Schenk et al. (2013); van Biljon et al. (2015); Johs-Artisensi et al. (2020)), and meaningful activities (van Biljon et al. (2015); Johs-Artisensi et al. (2020)) for residents.

It is important to note that these studies have been conducted in specific areas or localities, limiting their generalizability to other geographical or cultural settings. This limitation arises from the diversity observed among different residences, including variations in organizational structures, professionals involved, and types of residents (McCabe et al. (2021)). Additionally, there have been discussions surrounding the methods used to obtain information on QoL. To address these concerns, more specific cognitive interviewing techniques are proposed, particularly for vulnerable groups like older adults, in an attempt to mitigate the positive response bias often observed in NH settings (Meyer et al. (2019)).

The integration of newly identified dimensions, along with the established ones, contributes to a better definition of the QoL model (Carcavilla González et al. (2021)). To enhance the understanding of QoL, it is important to expand the range of techniques beyond questionnaires and interviews. Stakeholder assessments and other observational techniques can enrich the QoL dimension and help mitigate biases (Meyer et al. (2019)). Adapting general QoL measures to the NH context requires a deeper understanding of these dimensions. However, ongoing studies in this direction have identified a subset of factors from these scales that are relevant in the NH setting. These factors should be further expanded in models that account for residents' and NH-specific factors (Duan et al. (2021)).

Understanding the relationships between QoL variables and factors allows us to determine their importance and potential redundancy, as well as identify and integrate new variables. Within the group of eleven studies focused on identifying factors contributing to QoL in NH and understanding their relationships, several key findings emerged. Social support (Scocco and Nassuato (2017); Nordin et al. (2017); Wu et al. (2018); Maenhout et al. (2020); McCabe et al. (2021)), physical health (Scocco and Nassuato (2017); Nordin et al. (2017); Duan et al. (2021)), and psychological well-being (Nordin et al. (2017); Kloos et al. (2019)) were highlighted as significant factors influencing QoL in NHs. In addition to these factors, reflecting on personalized factors leads to the concept of cultural change in NHs. This approach not only promotes personalized QoL but also drives significant improvements in NH configuration to enhance residents' QoL (Burack et al. (2012)).

This study provides a comprehensive review of the factors that contribute to QoL and WB in older adults living in NHs. However, there are several limitations that should be taken into account when interpreting the results. One of the main limitations of the study is its focus on a specific population, namely older adults living in NHs. While this population is of great interest due to the challenges they face in maintaining QoL and WB, the findings may not be generalizable to other populations, such as older adults living in the community or receiving care in other settings. Therefore, caution should be exercised in generalizing the findings to the broader population of older adults or other contexts. Nevertheless, these studies have made significant progress in understanding how older adults perceive and experience various domains of QoL in NH facilities, providing valuable insights for interventions, resource allocation, and future research (van Biljon et al. (2015)). The findings of this review have revealed conceptual models of QoL specific to certain groups of older

adults in specific contexts. These models should be considered as approximations of social reality rather than comprehensive and detailed descriptions and representations.

Another limitation of this study is the search strategy used to identify relevant studies. The utilization of the PICo tool to define the search criteria may have restricted the breadth of the search and potentially excluded relevant studies that did not align with the PICo criteria. Additionally, only studies published in English or Spanish and those published in relevant journals or conferences were included, potentially resulting in the exclusion of relevant studies published in other languages or different types of publications. The study selection process was conducted by two authors with discrepancies resolved by a third author, which introduces some subjectivity into the selection process. Furthermore, while the data extraction process employed a standardized form, it is possible that some pertinent information may have been overlooked. These limitations may have impacted the comprehensiveness of the review and the accuracy of the findings. Nonetheless, this study provides valuable insights into the factors contributing to QoL and WB in older adults living in NH.

Finally, the comparability of the results reported in the described studies is limited due to the different methodological approaches employed. Older adults who declined or were unable to participate in the studies due to physical or psychological disabilities may possess distinct characteristics, leading to potential biases (Burack et al. (2012); Wu et al. (2018)).

The present study conducted a comprehensive and systematic literature search, covering numerous databases over an extended period of time, in which no systematic reviews on QoL in NHs were identified. The study highlights the current state of research in this area, providing an opportunity to reflect on the existing research directions. The aim is to integrate these findings into an advanced model in a near future tailored not only to NH settings but also to the specific needs and preferences of the residents themselves (Burack et al. (2012); Duan et al. (2021)).

## 5. Conclusions

Defining QoL is a complex task, as it depends on multiple factors related to the residents, their environment (Bowling et al. (2013); Nordin et al. (2017); Haugan et al. (2020)), family and social aspects (Straker et al. (2011); Godin et al. (2015); Roberts and Ishler (2018); Duan et al. (2021)), and even the geographical context (Haugan et al. (2020); Laybourne et al. (2021)), among others. It is crucial to develop a measurement approach that considers the different dimensions of QoL and can be validated in larger environments than those previously studied. These studies contribute to establishing a standardized definition of QoL and bring us closer to developing a specific QoL model for older adults living in NHs. This model should integrate different variables and measures, including the health, psychological, and social aspects of the residents, to ensure comprehensive QoL assessment for this population. Therefore, further research in this field is warranted.

This study presents a comprehensive analysis of the QoL and WB of older adults in NHs, emphasizing the importance of investigating the dimensions of QoL and developing an integrated model that examines their interrelationships. This research aims to enhance the QoL for residents in NHs. The review also identifies factors that influence QoL in NH, explores their relationships, and identifies potential predictor variables, providing valuable insights for interventions and policies aimed at improving QoL in this context. Overall, this research contributes to the understanding of QoL in NHs and lays the foundation for future studies in this field, ultimately leading to improved QoL and WB for older adults in NHs.

The models that have been developed and validated for specific populations and specific geographical areas in NHs remain relevant for those particular contexts. It is crucial to customize QoL questionnaires by considering specific domains that encompass various aspects of residents' lives. This integrative model of QoL serves as a starting point towards a personalized and comprehensive model that takes into account individual circumstances.

A tailored QoL model should be developed for residents that appropriately considers the multiple factors associated with their specific context and can be customized accordingly. In order to achieve this, it is important to consider NH culture change (Duan et al. (2021)).

Advancements in this area can contribute to the development of a personalized model of QoL for this vulnerable group. This study provides valuable insights not only for the staff directly involved in the care of older adults, such as nurses, but also for managers and administrators in improving the QoL of residents by offering new perspectives. Researchers can utilize this systematic review as a reference for future studies and the creation of enhanced QoL models.

**Author Contributions:** Conceptualization, A.R.-M. and Y.M.D.-l.-F.-R.; methodology, M.d.C.M.-C. and J.J.J.-D.; software, J.J.J.-D.; validation, A.R.-M., Y.M.D.-l.-F.-R. and M.d.C.M.-C.; formal analysis, A.R.-M. and J.J.J.-D.; investigation, A.R.-M.; resources, Y.M.D.-l.-F.-R.; data curation, A.R.-M.; writing—original draft preparation, A.R.-M.; writing—review and editing, Y.M.D.-l.-F.-R., M.d.C.M.-C. and J.J.J.-D.; visualization, M.d.C.M.-C.; supervision, J.J.J.-D. All authors have read and agreed to the published version of the manuscript.

**Funding:** This research did not receive any specific grant from funding agencies in the public, commercial, or not-for-profit sectors.

**Institutional Review Board Statement:** Not applicable.

**Informed Consent Statement:** Not applicable.

**Data Availability Statement:** No underlying data were collected or produced in this study.

**Conflicts of Interest:** The authors declare that there is no conflict of interest regarding the publication of this paper.

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
