# Peer review of "Quality of Life and Well-Being of Older Adults in Nursing Homes: Systematic Review"

_socsci, doi:10.3390/socsci12070418_

Round 1

Reviewer 1 Report

The manuscript "Quality of Life and Well-Being of Older Adults in Nursing Homes: Systematic Review" aims to know what is the state of research in the last decade on the aspects, measures and factors that affect or help determine the quality of life and well-being of older adults in nursing homes. To do this, they conduct a systematic review of the literature and perform a content analysis of the 21 studies that meet the criteria they have established. The manuscript is interesting and covers important topics. However, some minor changes should be made before publication:

- On page 1, on lines 32 to 34 it says: WB exists in two dimensions, subjective and objective, as defined by the WHO Regional Office for Europe: “It comprises an individual’s experience of their life as well as a comparison of life circumstances with social norms and values”. The reference is missing and should be included, both in the text and in the References section

 -  In Table 2. Quality assessment criteria scores for quantitative methodologies, assessment criteria 5 (If interventional and random allocation was possible, was it described?) 6 (If interventional and blinding of investigators was possible, was it reported) and 7 (If interventional and blinding of subjects was possible, was it reported?) should be removed since the manuscript, in section 2.3. Study selection establishes that one of the exclusion criteria is “studies about interventions”.

 -  In lines 323 to 325 it says “Dimensions related to personal, organizational, or activity-related factors, as well as social satisfaction can be used to determine their relationship with residents' QoL as stated in (Maenhout et al., 2020).” It would be more appropriate to put: “Dimensions related to personal, organizational, or activity-related factors, as well as social satisfaction can be used to determine their relationship with residents' QoL as stated in Maenhout et al. (2020).”

 -    And the same happens in lines 365-366 where it says: “Residents' perceptions of adequacy of care and having a reason for living in NH are predictors of QoL according to (Pramesona & Taneepanichskul, 2018)”. It would be more appropriate to put: “Residents' perceptions of adequacy of care and having a reason for living in NH are predictors of QoL according to Pramesona and Taneepanichskul (2018)”

Reviewer 2 Report

The manuscript focuses on an important topic, has a mostly clear review protocol, and draws upon a full body of literature. 

I have several suggestions to strengthen this manuscript before it is ready for publication. They are as follows:

- The introduction requires reorganization for flow and clarity. The first paragraph should follow an overall description of quality of life in nursing homes-- something that hooks the reader and grounds your aims/research question.

-As it stands, the research questions/aims are both too broad in scope and unclearly articulated. The research questions must be revised and narrowed, and methods, PRISMA table, and discussion should be revised to reflect these revisions. I implore authors to rebrand this review as a replication of Lee et al., 2009 to resolve some of the clarity/scope issues throughout the manuscript. Justify this replication beyond just timeframe.

-It is currently unclear if your main interest is in the actual measures of QOL/WB or beyond. It is possible that you need a manuscript solely for the measures and another manuscript for the other content you are analyzing.

-Nursing home needs to be operationalized. What is a nursing home-- is there a global definition? If not, what variations in culture, policy, etc are relevant to your methods? You touch on this in the limitations paragraph but needs to be frontloaded.

-Article selection process seems sound. I feel the lack of specificity in the RQ results in some confusion about what information should be displayed in Table 4. As a reader, I continue to lean toward feeling this could be two separate manuscripts. It is not always clear what the relationship between evaluation of QOL/WB measures and qualitative topics covered in results section.

- Throughout the discussion and conclusion, there are terms used without explanation or description (e.g., culture change). There needs to be more explicit links between findings, existing empirical literature, and suggestions for future research/practice.

-You discuss comparability as a limitation, but I think you need to attend to variables you bring up in your PRISMA (e.g., study population, methods of collection), as they are available for your analysis.

-Discussion and conclusion do not make a compelling case for why this research is beneficial, as it stands. What important contributions are made by your findings and how will they affect future research on QOL in nursing homes?

-Consider APA guidelines on usage of terms such as "elderly" or "seniors"-- gerontology no longer supports the use of this language to describe older populations. You may include them in search terms strategically, but they are no longer appropriate for use in scientific writing.

-Clarity, flow, and word choice can be improved with another proofreader or access to writing resources at one's university or prospective publisher.
